# The Current Therapeutic Role of Chromatin Remodeling for the Prognosis and Treatment of Heart Failure

**DOI:** 10.3390/biomedicines11020579

**Published:** 2023-02-16

**Authors:** Lindsay Kraus, Brianna Beavens

**Affiliations:** College of Science, Technology, Engineering, Arts and Mathematics, Alvernia University, Reading, PA 19607, USA

**Keywords:** cardiovascular disease, chromatin remodeling, heart failure

## Abstract

Cardiovascular diseases are a major cause of death globally, with no cure to date. Many interventions have been studied and suggested, of which epigenetics and chromatin remodeling have been the most promising. Over the last decade, major advancements have been made in the field of chromatin remodeling, particularly for the treatment of heart failure, because of innovations in bioinformatics and gene therapy. Specifically, understanding changes to the chromatin architecture have been shown to alter cardiac disease progression via variations in genomic sequencing, targeting cardiac genes, using RNA molecules, and utilizing chromatin remodeler complexes. By understanding these chromatin remodeling mechanisms in an injured heart, treatments for heart failure have been suggested through individualized pharmaceutical interventions as well as biomarkers for major disease states. By understanding the current roles of chromatin remodeling in heart failure, a potential therapeutic approach may be discovered in the future.

## 1. Introduction

Cardiovascular diseases (CVDs) have been the leading cause of death worldwide for many years [1,2]. It has been estimated that around 17.9 million lives are lost annually due to CVDs across the globe [2]. Currently, there are no cures for any type of CVDs; there are only preventative measures to alleviate risk factors and in extreme cases perform heart transplants [3,4]. CVDs include a multitude of pathological conditions, including atherosclerosis, ischemic heart disease, stroke, and heart failure, to name a few [5]. For this review, heart failure (HF) will be the focus. HF occurs due to structural or functional stress in the heart which no longer allows the heart to properly pump blood and oxygen to designated areas of the body [6,7,8]. This is often manifested in decreased cardiac output and increased internal cardiac pressure [7]. Annually, there are over 64 million cases of HF, accounting for over 346 billion US dollars, and these numbers are projected to rise [9]. It is predicted that the rate of HF will increase by 50% in low and middle sociodemographic regions by 2030 [9]. With no cure and increasing concern across the globe, there is a dire need to understand and treat this devastating disease.

Because of the lack of a cure for HF, many interventions have been utilized to attenuate risk factors. There is an assembly of drug-based interventions that have proven to be instrumental in treating the risk factors associated with HF, one of the most significant being chronic hypertension [10]. Heart transplants do occur, but not as frequently as they are needed [11]. To truly cure an injured heart, cardiac stem cells have been suggested and are currently being studied [12]. However, they have not been as successful as originally proposed due to engraftment issues and detrimental immune responses [13]. Alternatively, cardiac remodeling via epigenetic regulation has been studied and has had promising outcomes [14]. In recent research, the combination of epigenetic regulated chromatin remodeling methods with the techniques, advancements, and theories of stem cell regulation have been groundbreaking for the field of cardiac biology [15,16].

The role of chromatin remodeling in HF is not a new idea, partially due to the overwhelming application of epigenetics in cardiac biology [14,17,18,19]. Generally, it is believed that when epigenetic modifications are altered there is a progression or suppression of the cardiac disease state [20]. Normally, eukaryotic DNA is tightly condensed in chromatin, which is compacted around histones that form a nucleosome [21]. These histones have all been identified (H1, H2A, H2B, H3, H4) and are well understood in cardiac biology [22,23,24]. However, changes to these histones and the chromatin are less understood. Modifications such as a methylation and phosphorylation can be added to or removed from these structures, altering the chromatin architecture as well as the recruitment of complexes that can alter the overall chromatin structure [25]. These alterations dictate whether the DNA becomes more or less tightly condensed. The more compacted the DNA, the less available the genes, meaning less gene expression or transcription. If the chromatin is less condensed, the DNA is open and therefore the genes are more accessible and can allow for more gene expression and transcription [25]. Many of these mechanisms have been connected to the progression or suppression of genes involved in HF [17]. Specifically, chromatin remodeling has been the highlight of many investigations involving the development of the heart [23,24] as well as the disease state [25]. Chromatin remodeling is of specific importance because the architecture and rearrangement of the chromatin can dictate more than just one epigenetic site or change. Chromatin remodeling affects the accessibility of multiple transcription sites and the overall availability of the chromatin [26,27,28].

Targeting the changes in chromatin architecture or chromatin remodeling has been of interest over the past few years. With promising strategies, potential clinical trials, and the advancement of therapeutic approaches, chromatin remodeling in HF is a vital method for understanding this devasting disease to find a therapeutic approach to treating HF as depicted in Figure 1. The goal of this review is to summarize the most recent advancement in chromatin remodeling in the progression and treatment of HF.

## 2. Current Strategies for Detecting Chromatin Remodeling in Heart Failure

With advancements in bioinformatics as well as gene therapy technology, the ability to study chromatin as a clinical cardiac therapeutic has greatly increased over the last decade [29,30,31,32,33]. Because chromatin is often tightly compacted and densely organized in the nucleus of cells, it can be difficult to isolate [34]. Importantly, variations to the chromatin architecture are a three-dimensional, multifaceted challenge for researchers [18]. To truly find a therapeutic approach, it is important to know the current challenges and new advancements in studying chromatin remodeling in the heart. Currently, there are standard approaches to studying changes in chromatin remodeling and gene expression due to changes in epigenetic profiles [35,36,37]. Many of these approaches include a combination of genomics, RNA molecules, cardiac gene targets, and chromatin remodelers. Advancements in bioinformatics and gene therapy have proven to be pivotal in understanding pathological signaling in cardiac disease states; however, this can prove challenging and too broad for a true therapeutic approach. Additionally, targeting only a specific change in selective gene expression can be difficult and can cause other adverse downstream effects as well [38]. However, a combination of all of these advancements could provide information to build a comprehensive understanding of chromatin remodeling during HF.

A more recent and promising strategy is using small molecules to target chromatin remodeling in the heart. By using chromatin remodeling enzymes, histone modifiers, chromatin regulatory complexes, and even DNA modifiers, the changes in chromatin architecture can be directly studied. This has been shown with the Switch/Sucrose-Nonfermentable (SWI/SNF) chromatin remodeling complex in many oncological studies, but it could also prove valuable in the heart [39]. Similarly, small molecules such as long noncoding RNAs (lncRNAs) have been used to assess chromatin remodeling [40]. The lncRNAs have been directly associated with structural changes in the chromatin in many disease models, including during pathological challenges to the heart [28,41]. One of the biggest advancements for understanding chromatin remodeling during HF is through genetic profiling approaches via sequencing and bioinformatic-based analysis. Using ATAC sequencing, RNA sequencing, and ChIP sequencing has led to major advancements in understanding epigenetic changes during HF. Specifically, for chromatin remodeling, combining these approaches in a novel “multiomic” approach has led to major discoveries that will be addressed in this review.

## 3. Therapeutic Approaches Using Chromatin Remodeling in Heart Failure

### 3.1. Genomics and Bioinformatics

The use of genomics to understand chromatin alterations and all the downstream changes has become a steadfast approach to most clinical applications of HF research. Many genomics-based research techniques are incorporated into most current studies along with other techniques, as this review will highlight. It is important to emphasize some of the most influential sequencing techniques that have made an impact on the field of chromatin remodeling in cardiac biology. One of the main methods to study chromatin remodeling through genomics is through the Assay for Transposase Accessible Chromatin (ATAC) sequencing. This ATAC sequencing data allows researchers to understand chromatin remodeling, chromatin accessibility, and epigenetic profiles [37]. Through this sequencing method, studies have found pivotal transcriptional and chromatin accessibility differences in HF [42]. For example, one study used ATAC sequencing to understand trans-aortic constriction in mice and found predictors of chromatin structural changes that led to a pivotal understanding of gene expression changes during disease progression [43]. Another study used ATAC sequencing to analyze different genes found in hypoxia-induced stress associated with HF and found major changes in the chromatin’s accessibility during stress [44]. ATAC sequencing is often paired with other genomic techniques or gene expression data.

Many of the widely used techniques include ATAC sequencing, RNA sequencing, miRNA sequencing, and DNA sequencing. All of these are not new techniques; however, how the data is used and interpreted is constantly changing. For example, one study used the multiomic approach, meaning they used single-cell RNA sequencing (scRNA seq), single-cell ATAC (scATAC seq), bulk ATAC sequencing, and miRNA sequencing to study HF in murine models [45]. From these immense data sets, the authors found miRNA expression differences that aligned with chromatin accessibility profiles in mice with HF that were not seen in the control mice. All of this gave a new understanding of potential targets and mechanisms in the progression of HF [45]. Another study used the multiomic approach to map changes in cardiomyocytes during differentiation and found a specific connection to the ACTN2 gene that was not known previously [46]. Using DNA sequencing, a study performed genome-wide chromatin conformation capture (Hi-C) to study adult cardiomyocytes in HF [47]. This study focused on the loss of a chromatin structural protein, CTCF. The loss of CTCF led to increased HF in mice and structural differences in the epigenomic profile. Specifically, it was determined that the loss of CTCF led to a loss of chromatin interaction specifically around pathologically related cardiac genes [47]. Similarly, genomics has also been suggested and utilized as a biomarker for HF [48]. One study assessed 111 families with various types of heart disease and used data sets generated via whole genome sequencing. They found that sequencing could be used as an initial understanding of human based cardiac diseases [49]. Again, genomic-based research techniques are a necessary component of research used for clinical relevance; however, genomics only provides one piece of a very complex puzzle.

### 3.2. Targeting Cardiac Genes for Transcriptional Regulation

Chromatin remodeling in the heart is regulated by a multitude of genes. Some of these cardiac genes have been targeted during HF to assess changes in the cardiac disease state. For example, the transcription factor, Med1, was found to regulate chromatin remodeling in cardiomyocytes. Specifically, Med1 was found to synchronize the histone acetylation of lysine 27 (H3K27) which allowed for more open chromatin accessibility and therefore more gene expression [50]. One study found that changes in gene expression in GATA4 and NXK2.5 were associated with changes in the chromatin architecture [43]. Interestingly, the authors found a decrease in the protein CTCF, which is a chromatin structural protein, specifically after HF in murine cardiomyocytes [43]. Likewise, another study found that GATA4 plays a major role in the chromatin structure, specifically in cardiac disease progression as well as cardiac development [51]. He et al. describe these changes as GATA4 occupancy, which is significantly altered during fetal cardiac development. Similarly, they saw changes in gene expression and therefore chromatin occupancy during cardiac stress [51]. The transcription factor, AP-1, was found to play a pivotal role in cardiomyocyte regeneration and growth after injury in zebrafish [52]. Zebrafish are known to have an increased ability to regenerate and grow cardiomyocytes even into adulthood, unlike humans and mice, making them an interesting resource for studying HF [53]. Thus, this study connected AP-1 to the regulation of chromatin remodeling and accessibility in the heart, making it a viable target for cardiomyocyte regeneration after injury [52].

It has also been found that changes in chromatin architecture have a role in cardiac fibroblast phenotypic gene expression. After cardiac injury, there is often an influx of cardiac fibroblasts that form scar tissue in the heart. Too much fibrosis can lead to irreversible damage and decreased heart function [54]. It has been suggested that transcription factors such as GATA4, as well as MEF2C and TBX5, could be directly reprogrammed to alter their chromatin remodeling and therefore epigenetic signaling in fibrosis [52,53]. This has specifically been studied in cardiac fibroblasts in vitro and in vivo with high success [55,56,57,58,59]. Additionally, changes in chromatic structure were shown to affect the stiffness of cardiac fibroblasts after injury [60]. It was also found that topological change to cardiac fibroblasts significantly altered the chromatin remodeling and downstream cardiac gene expression associated with a decreased pathological response [61]. The chromatin remodeling protein, BRG1, was also found to be a vital regulator of cardiac fibrosis, specifically in regulating the endothelial-to-mesenchymal transition [62].

Chromatin remodeling has also been connected to metabolism during HF. Specifically, one study found that changes in the expression of beta hydroxybutyrate (BHB) were associated with alterations in chromatin remodeling and histone modifications that enhanced disease progression [63]. Chromatin remodeling and BHB expression were associated with cardiac protection via mitochondrial function and metabolism [63]. Overall, an important conclusion from assessing gene regulation is that changes in gene expression alone do not tell the entire story. There is a combination of regulation from gene expression, modifiers, and epigenetic marks that is necessary to fully understand chromatin remodeling in the heart, which is summarized in Table 1.

### 3.3. RNA Molecules

The use of small RNA molecules, often deemed non-coding RNA molecules, has been of great interest in epigenetic regulation and chromatin remodeling, specifically in HF [64]. Many RNA molecules have been studied in pathological cardiac models, microRNAs (miRNAs), circular RNA (circRNA), and long non-coding RNAs (lncRNAs) being the primary focus due to influential and novel data [65,66,67,68].

It has been shown that the lncRNAs are regulated by ATP-dependent chromatin remodeling factors in both mouse and human hearts during HF [40]. The lncRNAs have also been used as biomarkers for HF [65,66]. Current research has indicated that these lncRNAs may be the master regulators of disease progression in HF via chromatin remodeling [69,70,71]. Many lncRNA can bind and regulate chromatin remodeling through epigenetic modifications and therefore regulate various cardiac gene expression levels during pathological stress [61,68]. For example, an lncRNA labeled AHIT was found to regulate cardiac hypertrophy by binding to SUZ12, which is a part of the polycomb repressive complex 2 (PRC2), known to regulate histone modifications and chromatin remodeling [72]. Another study found that the lncRNA NEAT1 increased cardiac fibrosis through EZH2, which is an epigenetic modifier known for cardiac gene repression [73]. Interestingly, the lncRNA CHAIR was found to be cardioprotective by modulating DNA methylation. Specifically, CHAIR inhibited the DNA binding complex, which limited the pathological response in the heart [74]. Finally, the lncRNA labeled MEG3 was found to regulate chromatin remodeling, which specifically induced matrix metalloproteinase 2 (MMP2), increasing cardiac fibrosis [75]. The role of lncRNAs in cardiac chromatin remodeling is an expanding source of innovative research for the treatment of HF.

The circular RNAs (circRNA) have also been of interest due to their ability to act as a sponge as well as a transcriptional regulator throughout cardiac tissue [76]. Because of their unique role in cardiac tissue, circRNAs have been proposed as major biomarkers for HF [77]. For example, one study found that the circRNA named circNCX1 played a major role in cardiomyocyte death during cardiac injury. Specifically, there was an increase in this circNCX1 with an increase in reactive oxygen species (ROS). The knockdown of this circRNA decreased the amount of cardiomyocyte cell death in mouse hearts [78]. Often circRNAs do not act completely alone. One study found that the circRNA circ-HIPK3 interacted with the miRNA miR-17-3p to regulate calcium signaling during HF. Specifically, the downregulation of the circRNA seemed to lessen the fibrotic response and the progression of HF in adult mice [79].

Another major RNA molecule studied in cardiac biology is the miRNAs, known for extensive signaling networks and chromatin regulation in the heart [80]. Because of the interconnected signaling, miRNAs are often connected to other RNA molecules and other signal transduction. One study found a connection between the miRNA miR-193b and ROS. Specifically, there was an increase in both the miR-193b and ROS in HF connected to a metabolic syndrome that increased pulmonary dysfunction in exercise-induced pulmonary hypertension [81]. The gene NRF2 has been associated with antioxidant redox balance that often becomes dysregulated in HF. Many myocardial miRNAs have been associated with the dysregulation of NRF2 during HF [82]. Overall, RNA molecules have great potential for unlocking biomarkers and signaling associated with chromatin remodeling during HF, which could provide an invaluable resource with therapeutic potential.

### 3.4. Chromatin Remodelers and Complexes

Unlike RNA molecules and cardiac gene transcription, chromatin remodelers play a direct role in regulating chromatin architecture. When the chromatin remodelers are dysregulated, the chromatin structure is altered, which can lead to devasting pathological responses that cause increased cardiac cell death, inflammatory signaling, and stress responses found in HF [28]. One study found that the chromatin remodeling protein BRG1 and p300 had stage-specific regulation of histone acetylation. Specifically, these chromatin remodelers were upregulated during HF but not during left ventricular hypertrophy, indicating a step-wise transition regulated by chromatin regulators [83]. The chromatin regulator, identified as SETD7, known for the methylation of histone 3 at lysine 4 (H3K4me1), was found to regulate inflammation pathways in obese patients with HF [84]. It was found that the loss of SETD7 in murine cardiomyocytes protected against hypertrophy and further cardiac dysfunction that was directly associated with the regulation of inflammatory genes [84]. Another study found that the interaction between a histone lysine methyltransferase named G9a and its downstream target Brain Derived Neurotrophic Factor (BDNF) regulated histone epigenetic modifications, therefore altering chromatin remodeling. It was found that G9a, which inhibits BDNF and increases cardiomyocyte death, is overexpressed in HF [85]. Another study found that ZNHIT1, a major regulator of a chromatin remodeling complex, was necessary for heart function [86]. Specifically, the loss of this chromatin regulator caused rapid HF and dysregulated calcium signaling. It was determined that the ZNHIT1 regulated CASQ1, a major regulator of calcium signaling in the sarcoplasmic reticulum, by altering the histone 2A variant and therefore chromatin regulation [86]. It was also found that BAF60c, a chromatin remodeling complex also called SMARCD3, regulates cardiomyocyte growth through MEF2 and myocardin gene expression [87].

Two major chromatin remodeling complexes are histone deacetylases (HDACs) and histone acetyltransferases (HATs). Both of these are known to regulate lysine acetylation in many cell types, including cardiac cells [88,89]. The dysregulation of both HATs and HDACs has been associated with increased HF phenotypes and cardiac disease progression [90]. Pharmaceutical inhibitors of these complexes have been suggested as a therapeutic approach to cardiac diseases [91]. Many studies have connected HDAC inhibition with lessening cardiac fibrosis seen in HF [92,93,94]. This HDAC inhibition has directly improved heart function [95], making it a major therapeutic target for HF. Additionally, HDAC regulation has been associated with DNA damage mechanisms associated with aging. The SIRT1, a class III HDAC, was found to regulate cellular senescence in cardiac cells, which was associated with increased stress and cardiac injury [27].

It has also been well studied that the bromodomain and extraterminal (BET) chromatin regulators, such as BRD4, play a major role in maintaining cardiomyocyte health, specifically via mitochondrial function, during HF [96]. Other studies have seen similar advancements and chromatin remodeling therapeutic potential using BET regulation in HF [92,93,97]. One study found that BET inhibitors (BETi) caused increased DNA damage responses and altered overall chromatin structure, resulting in better cell viability. This was connected to BRD4 colocalizing with YY1 at a Topologically Associating Domain Boundary (TADB) [98]. Similarly to HDACs, BET inhibition has been a major topic for drug-based chromatin remodeling interventions.

As more and more chromatin remodeling complexes have been uncovered, the role of super-enhancers within those complexes has been discovered as well. Super-enhancers are often areas in the genome that have an increased number of chromatin interactions [99]. These super-enhancers have been studied regarding chromatin remodeling during HF as a major target for cardiac disease progression inhibition [100]. Likewise, it has been studied that ATP-dependent chromatin remodeling complexes are extremely important in regulating pro-inflammatory responses, making them interesting targets for biomarkers of hypertension during HF [101,102]. There have also been connections made between chromatin remodelers and global chromatin architectural changes, specifically ones associated with heart failure and aging mechanisms. Chromatin remodeling dysregulation was specifically connected with skeletal muscle changes in aging models [103]. Overall, chromatin remodelers are a promising strategy for uncovering epigenetic modulation during HF.

**Table 1 biomedicines-11-00579-t001:** Summary of chromatin remodeling mechanisms during heart failure.

Chromatin Alterations	Summary of Function	Source
Target Genes Med1 Gata4 Nxk2.5 CTCF AP-1 MEF2c Tbx5 BRG1 BHB	Transcriptional regulation, caridac development, metabolism, cardiac fibrosis, cell death regulation, epigenetic regulation, chromatin remodeling	[51,52,53,54,55,56,57,58,59,60,61,62,63]
RNA Molecules lncRNAs AHIT CHAIR MEG3 circRNA circNCX1 circHIPK3 miRNA miR-17-3p miR-193b	Regulating epigenetic marks, cardiac gene expression, chromatin remodeling, apoptoic regulation, redox regulation	[64,65,66,67,68,70,72,73,74,75,76,77,78,79,80,81]
Chromatin Remodelers BRG1 P300 SETD7 G9a Znhit1 Baf60c HDACs HATs BETs/BRD4	Direct chromatin architecture regulation, adding and removing epigenetic modifications, cardiac disease regulation, DNA damage respones, pro-inflammatory response regulation, aging mechanisms	[28,83,84,85,86,87,88,89,90,91,92,93,94,95,96,98,99,100,101]

## 4. Current Clinical Application of Chromatin Remodeling for the Treatment of Heart Failure

Because of the influential and groundbreaking research on chromatin remodeling in HF, some research has progressed into clinical applications as a form of treatment for the disease. Many have suggested using epigenetic modifications and chromatin remodeling dysregulation as biomarkers for HF [102]. More recently, the use of the epitranscriptiome, which consists of RNA epigenetic coding, has been suggested. This incorporates different RNA molecules and further gene expression [89,90]. One study suggested the “EPi-transgeneratIonal networK mOdeling-STratificatiOn of heaRt Morbidity” (EPIKO-STORM) platform, which helps log the history of HF. The suggested application of it is to document the epigenetic history of patients to help better predict familial related HF risk factors that are not as common [104]. As described earlier, the use of multiple omics, or multiomic, approaches has also been applied to clinical applications [104,105,106,107]. There is also a great benefit to using genomics-based approaches to diagnose any hereditary based cardiac diseases [31]. Through a combination of many different omics approaches, a more comprehensive understanding is continuously developing to uncover the genetic and epigenetic mechanisms of HF in patients.

Because of the lack of a cure for HF currently, individualized medicine is at the forefront of clinical applications [108]. This has led to “epi-drugs” or epigenetic-based pharmaceutic interventions for patients with HF [94,95]. It is thought that epi-drugs could alter a specific epigenetic profile by adding or removing an epigenetic mark and therefore alter chromatin remodeling in a pathological setting. Interestingly, these epi-drugs have been approved for other diseases [109], specifically some cancers [110]. Some epi-drugs that have been considered and studied included targeting the chromatin remodeling complexes. For example, it has been shown that HDAC inhibitors can be used to help lessen inflammatory responses, pathological gene expression, and fibrosis during HF [111,112,113,114,115]. Clinical applications are looking at DNA methylation modifiers [116,117], BET inhibitors [118], and even chromatin remodeling miRNAs [102,111]. Similarly, it has been suggested that interventions, such as the epi-drugs, could be used to target transcriptional-based machinery use in chromatin remodeling [118]. Finally, there is evidence to suggest that altering the gut microbiota can even change chromatin remodeling in cardiac related genes [108,119,120]. It was specifically found that HDAC3 activity could be altered with microbiota-based metabolites [119]. Overall, there are a variety of promising therapeutic techniques being studied and developed for HF interventions using chromatin remodeling.

## 5. Conclusions

Chromatin remodeling is a promising strategy not only for understanding HF but also for potentially treating it. Because of the influential role of the genomic architecture and the regulation of gene expression, targeting chromatin remodeling is an encouraging approach for altering cardiac genes during stress or injury. However, with all the great advancements, there are still major hurdles to overcome. Additionally, the scope of this review focused on chromatin remodeling. It is important to address that epigenetic regulation in cardiac disease progression encompasses more than just chromatin remodeling, though this is not addressed specifically in this review. Many of the approaches have limitations that still need to be addressed, including massive genomics-based interventions, which can overlook details and create correlations that are not necessarily connected. Pharmaceutical-based chromatin remodeling interventions, such as RNA molecules and epi-drugs, have the advantage of being less invasive but still need to address adverse effects, the timing of administration, and the specificity [108,121]. In the future, using the techniques as biomarkers is one of the most promising applications, and with more regulation, consistency, and standardization, they could be pivotal in addressing HF in patients before it becomes deadly [121,122,123]. As therapeutic advancements are made, the use of chromatin remodeling as a pharmaceutical intervention and as a biomarker for cardiac disease states would be monumental for heart failure patients. Overall, chromatin remodeling is an exciting and encouraging approach to HF that continues to grow to eventually cure this disease.

## Figures and Tables

**Figure 1 biomedicines-11-00579-f001:**
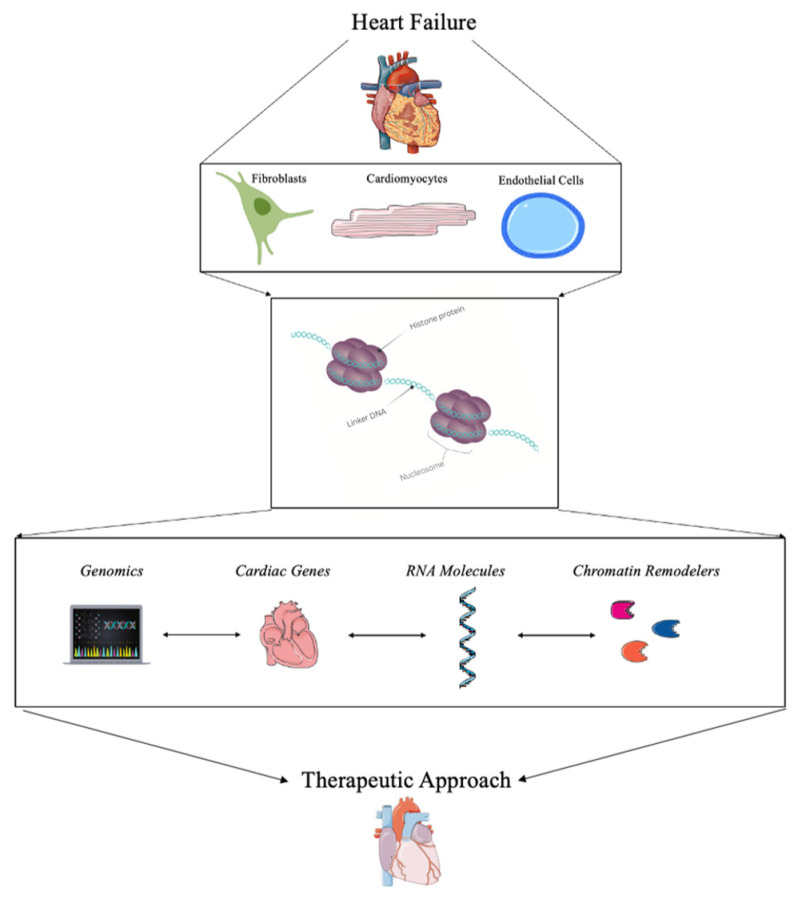
Summary of the role of chromatin remodeling for the treatment of heart failure.

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
