# Peer review of "The Current Therapeutic Role of Chromatin Remodeling for the Prognosis and Treatment of Heart Failure"

_biomedicines, 2023, doi:10.3390/biomedicines11020579_

Round 1
Reviewer 1 Report
This is generally a well-structured, well-written and comprehensive article, the data are presented in an appropriate manner, being clear and transparent. I consider that the study is valuable and sound and can be published after some minor revisions.
1. Please introduce a table and a figure significant for this topic.
2. I suggest addressing the future scope and topics that are important and that could not be covered in the manuscript.
3. The limits of this study should be well emphasized.
Author Response
We would like to thank the Reviewer for their time and comments. All of the suggestions have been incorporated into the manuscript. A summary figure is now present in the manuscript that encompasses the role of chromatin organization and remodeling in the context of heart failure treatment. Additionally, a summary table has been incorporated as well. The future scope and topics that were not addressed are now mentioned in the concluding paragraphs of the manuscript. The limitations are also more well-defined and addressed in the conclusion section.
Reviewer 2 Report
The review by Kraus and Beavens is interesting and thought provoking. There is a lot of content to the review which is useful for the community.
I have a couple of suggestions:
(1) Include some figure/diagram of chromatin organisation/nucleosomes/remodelling. This will be useful for non-specialists.
(2) I am interested if anything is known about genomic instability and DNA repair within the heart? Does chromatin remodelling/organisation have a role within those processes in relation to disease?
Author Response
We would like to thank the Reviewer for their time and comments. All of the suggestions have been incorporated into the manuscript. A summary figure is now present in the manuscript that encompasses the role of chromatin organization and remodeling in the context of heart failure treatment. Additionally, a summary table has been incorporated as well. We also added context to the genomic instability and DNA repair relative to chromatin remodeling during heart failure. Interestingly, it has been strongly associated with some HDACs, specifically, class III HDACs have been associated with DNA repair mechanism regulation via chromatin remodeling that is correlated with aging in the heart. This has been added to the section and the table.
Reviewer 3 Report
In this Review the Authors focuses on role of chromatin remodeling for the prognosis and treatment of heart diseases and failure. Using extensive analysis of the literature, they highlighted current strategies for detecting chromatin remodeling, main chromatin related therapeutic approaches, and current clinical application of chromatin remodeling for the treatment of heart failure.
This Review is good written and presents novel and important evidence. It can be published. There is only one small recommendation:
It would be good to discuss the effects of global epigenetic changes of chromatin architecture associated, for example, with polyploidy or laminopathy (10.3390/ijms23179691). If it is possible, please, describe in more detail the association between oxidative stress and chromatin remodeling.
In conclusion, please, underline novel findings provided by the Review.
Author Response
We would like to thank the Reviewer for their time and comments. All of the suggestions have been incorporated into the manuscript. We have incorporated the global epigenetic changes of chromatin remodeling in relation to oxidate stress associated with the aging mechanism in heart failure. Additionally, the novel findings are now clearly addressed in the conclusion.